# Removal of Erythromycin from Water by Ibuprofen-Driven Pre-Organized Divinyl Sulfone Cross-Linked Dextrin

**DOI:** 10.3390/polym16081090

**Published:** 2024-04-13

**Authors:** Mariano Ortega-Muñoz, Sarah Alvarado, Alicia Megia-Fernandez, Fernando Hernandez-Mateo, Francisco Javier Lopez-Jaramillo, Francisco Santoyo-Gonzalez

**Affiliations:** 1Department of Organic Chemistry, Faculty of Sciences, University of Granada, 18073 Granada, Spain; 2Unit of Excellence in Chemistry Applied to Biomedicine and the Environment, University of Granada, 18073 Granada, Spain; 3Biotechnology Institute, University of Granada, 18071 Granada, Spain

**Keywords:** cross-linking, dextrin, divinyl sulfone, erythromycin, emerging pollutant, biodegradable polymers, sorbent material, water management

## Abstract

Water recycling and reuse are cornerstones of water management, which can be compromised by the presence of pollutants. Among these, pharmaceuticals can overcome standard water treatments and require sophisticated approaches to remove them. Sorption is an economically viable alternative limited by the need for sorbents with a sorption coefficient (Kd) higher than 500 L/kg. The cross-linking of dextrin (Dx) with divinyl sulfone (DVS) in the presence of 1 mmol or 5 mmol of ibuprofen (IBU) yields the insoluble polymers **pDx1** and **pDx5** with improved affinity for IBU and high selectivity towards erythromycin (ERY) and ERY Kd higher than 4 × 10^3^ L/kg, when tested against a cocktail of six drugs. Characterization of the polymers shows that both **pDx1** and **pDx5** have similar properties, fast sorption kinetics, and ERY Kd of 13.3 × 10^3^ for **pDx1** and 6.4 × 10^3^ for **pDx5**, representing 26.6 and 12.0 times the 500 L/kg threshold. The fact that new affinities and improvements in Kd can be achieved by cross-linking Dx in the presence of other molecules that promote pre-organization expands the applications of DVS cross-linked polysaccharides as sustainable, scalable, and environmentally friendly sorbents with a potential application in wastewater treatment plants (WTPs).

## 1. Introduction

Water management is a current challenge and water-stressed regions are spread across all continents. According to the UN World Water Development Report 2020, water and climate change are linked. This link poses a risk to water resources [1]. In addition, the water needs have grown more than twice the rate of population increase, compromising the delivery of reliable water services [2]. Water recycling and reuse are critical issues to address water scarcity, particularly in agriculture, which accounts for 70% of global freshwater withdrawals and more than 90% of its consumptive use [2].

The presence of pollutants can compromise water reuse and make treatment processes more costly. Among the various emerging pollutants, pharmaceuticals pose a challenge because wastewater treatment plants (WWTPs) are not designed to remove them [3]. Antibiotics are a major concern, with global consumption reaching 34.8 billion daily doses in 2015, including those used in farms [4]. Antibiotic consumption plays a central role in the spread of antibiotic-resistance genes (ARGs), which promote antibiotic-resistant bacteria (ARBs). Although antibiotics may not pose a direct risk to human health due to their low toxicity, the emergence of ARGs and ARBs is an evolutionary pressure that contributes to the development of antibiotic resistance (AR) in humans, which is a serious health threat. For example, the US Centers for Disease Control and Prevention (CDC) estimates that more than 2.8 million antibiotic-resistant infections occur in the US each year, resulting in more than 35,000 deaths [5]. In addition, AR causes poverty and has a direct impact in the healthcare system, with an estimated increase in global healthcare costs from USD 300 billion to more than USD 1 trillion per year by 2050 [6].

A recent study of AR in US streams affected by WWTP discharges under varying instream flow conditions found that under low instream flow conditions, 26% of the streams did not meet the AR safety threshold for ciprofloxacin (CIP) and erythromycin (ERY) [7]. It is important to note that 5% of the dose of ERY is excreted in the active form in the urine and that ERY is recalcitrant to different treatments in WWTPs, with wastewater from WWTPs, pharmaceutical facilities, and hospitals being the main contributors to the spread of ERY [8]. As a result, varying concentrations of ERY have been reported in inland waters, groundwaters, marine systems, biosolids and sewage sludge from WWTPs, and sediments and its presence in finished tap/drinking water and the bioaccumulation in aquatic biota is of particular concern [8]. 

The health implications of ERY resistance are significant. ERY is a macrolide antibiotic prescribed for the treatment of various gram-positive and gram-negative bacterial infections and as an alternative to penicillin in patients allergic to this antibiotic, as well as for veterinary use. ERY resistance has been reported in *Streptococcus pneumoniae*, *Streptococcus agalactiae* [5], *Neisseria gonorrhea* [9], *Bordetella pertussis* [10], and in meat-associated bacteria, among others [11]. A recent study on the ERY resistance in the blood of *Staphylococcus aureus*, an indicator organism of the European Antimicrobial Resistance Surveillance Network, reveals a correlation between ERY resistance in blood methicillin-susceptible *Staphylococcus aureus* and the consumption of macrolide, lincosamide, and streptogramin B antibiotics [12].

The removal of ERY from water has been addressed by the implementation of different methods including coagulation-flocculation, powered activated carbon, granular activated carbon, reverse osmosis, advanced oxidation processes, membrane bioreactors, or biological activated filters, with varying degree of success [13]. However, when considering their application, cost is a critical parameter to consider, particularly in less developed countries. We have previously demonstrated the hypothesis that the cross-linking of carbohydrates with divinyl sulfone (DVS) is a feasible approach to obtain biodegradable sorbent polymers from economically affordable and sustainable starting materials [14,15]. In the context of antibiotic removal, we have reported that the cross-linking of starch with DVS yields a low-cost and eco-friendly sorbent polymer. This polymer effectively traps CIP from water with a Kd of 1469 L/kg and removal rates exceeding 92% [16]. 

In this work, we further explore our hypothesis of using DVS cross-linked carbohydrates to obtain affordable insoluble sorbent materials as an alternative to more sophisticated approaches to treat wastewater. We report that the cross-linking of Dx in the presence of IBU yields polymers with a high affinity toward ERY. This strategy, based on cross-linking in the presence of other molecules that promote a pre-organization to improve Kd and yield new affinities, expands the potential application of cross-linked carbohydrates sorbent polymers to different pollutants.

## 2. Materials and Methods

### 2.1. Reagents

Dextrin from potato starch (Dx, Fluka, Saint Louis, MO, USA), divinyl sulfone (DVS, 99.5%, TCI, Zwijndrecht, Belgium), ibuprofen [(RS)-2-(4-Isobutylphenyl)propanoic acid] (IBU, 98%, BDLpharm, Kaiserslautern, Germany), erythromycin [(3R,4S,5S,6R,7R,9R,11R,12R,13S,14R)-6-{[(2S,3R,4S,6R)-4-(Dimethylamino)-3-hydroxy-6-methyltetrahydro-2H-pyran-2-yl]oxy}-14-ethyl-7,12,13-trihydroxy-4-{[(2R,4R,5S,6S)-5-hydroxy-4-methoxy-4,6-dimethyltetrahydro-2H-pyran-2-yl]oxy}-3,5,7,9,11,13-hexamethyloxacyclotetradecane-2,10-dione] (ERY, >98%, TCI, Zwijndrecht, Belgium), atenolol [(RS)-2-[4-[2-hydroxy-3-(1-methylethylamino)propoxy]phenyl]ethanamide] (98%, TCI, Zwijndrecht, Belgium), hydrochlorothiazide [6-chloro-1,1-dioxo-3,4-dihydro-2H-1,2,4-benzothiadiazine-7-sulfonamide] (>97%, TCI, Zwijndrecht, Belgium) ciprofloxacin [1-Cyclopropyl-6-fluoro-4-oxo-7-(piperazin-1-yl)-1,4-dihydroquinoline-3-carboxylic acid] (CIP, 98%, TCI, Zwijndrecht, Belgium), ofloxacin [(RS)-9-Fluoro-2,3-dihydro-3-methyl-10-(4-methylpiperazin-1-yl)-7-oxo-7H-pyrido[1,2,3-de]-1,4-benzoxazine-6-carboxylic acid] (98%, BDLpharm, Kaiserslautern, Germany), and carbamazepine [5H-Dibenzo[b,f]azepine-5-carboxamide] (98%, BDLpharm, Kaiserslautern, Germany) were used as received. Anhydrous sodium carbonate (99.5%) and anhydrous sodium acetate (99%) were purchased from Sigma-Aldrich (St. Louis, MO, USA).

### 2.2. Synthesis of DVS Cross-Linked Polymers

The pH of 1 mmol (0.206 g) or 5 mmol (1.03 g) of IBU in 200 mL of water was adjusted to 9.5 by adding 13 mL of 0.83 M sodium carbonate. Subsequently, Dx (10 g) was added and sonicated to promote the dissolution. After 5.5 h of gentle stirring, 100 mL of water and 5 mL of DVS (5.7 g, 48.4 mmol) were added. The stirring was then extended for 30 min prior to adding 187 mL of 0.83 M sodium carbonate to reach pH 12 (Appendix A). The reaction proceeded for 16 h with gentle stirring as maintenance. The resulting solid was isolated by filtration and washed thoroughly with water until a neutral pH was achieved. The remaining IBU was protonated by acidulating with 5% HCl. The mixture was then washed with methanol and finally with diethyl ether. After drying under vacuum for 18 h at 40 °C, the obtained polymer amounts were 7.86 g and 5.43 g for 1 mmol (**pDx1**) and 5 mmol (**pDx5**) of IBU, respectively. A control polymer (**pDx0**) was synthesized under the same conditions but in the absence of IBU, yielding 8.41 g.

### 2.3. Characterization

Polymers were characterized by elemental analysis with a Thermo Scientific Flash 2000 elemental analyzer (Thermo Scientific, Waltham, MA, USA) to determine the presence of S from the sulfone group of the DVS cross-linker. Structural characterization of the polymers was addressed by X-ray powder diffraction (XRPD) and Fourier transform spectroscopy (FTIR). X-ray diffractograms were collected using a D8 discover instrument equipped with a Pilatus3R 100K-A detector (Bruker, Billerica, MA, USA) and a Cu Kα sealed tube (λ = 1.54 Å). The operation voltage and current were set to 50 kV and 1 mA, respectively. Data were collected from 2θ 5° to 85° with a 0.02° step and 40 s of integration. IR spectra ranging from 400 to 4000 cm^−1^ were measured with a Spectrum Two FT-IR spectrometer (PerkinElmer, Waltham, MA, USA) in ATR mode by accumulating 30 scans. 

Polymers were subjected to electron microscopy to observe their morphology and to thermogravimetric analysis (TGA) to study their thermal stability and identify the products of decomposition. The samples underwent electron microscopy analysis using a Zeiss SUPRA40VP field emission scanning microscope (Zeiss, Cluj-Napoca, Romania) after being covered with carbon. TGA was performed in a nitrogen atmosphere at 950 °C with a heating rate of 20 °C/min using a Shimadzu TGA-50H instrument (Shimadzu, Kyoto, Japan) coupled to a Nicolet 550 IR-FT spectrometer (Thermo Scientific). 

### 2.4. Sorption Studies

Sorption experiments were conducted in triplicate in batch mode at room temperature. To study the sorption of IBU, the polymers (0.1 g) were mixed with 10 mL of IBU water solutions (with concentrations ranging from 0.5 to 2 mg/L) in Falcon tubes and shaken in a tube rotator (VWR) for 3 h. Next, the IBU solution was separated by centrifugation at 4000 rpm and quantified its concentration using an F2000 fluorescence spectrophotometer (Hitachi, Tokyo, Japan) by interpolating the emission at 290 nm (λex 260 nm) in a calibration curve [17].

For the evaluation of the polymer as sorbents of a cocktail of drugs, 0.1 g of polymer and 10 mL of a water solution containing a mixture of carbamazepine, atenolol, hydrochlorothiazide, ofloxacin, CIP, and ERY ranging from 2 to 50 µg/L of each drug was assayed as described above and analyzed by mass spectrometry. With the help of an Acquity FTN AutoSampler (Waters Corporation, Milford, MA, USA), a volume of 10 µL of sample was injected in an ultraperformance liquid chromatography system (UPLC) comprising a Quaternary Solvent Manager Acquity (Waters Corporation) equipped with a BEH C18 column (1.7 mm 100 mm) coupled to a triple quadrupole mass spectrometer XEVO-TQS (Waters Corporation). The mobile phase of the UPLC system consisted of 0.1% (*v*/*v*) formic acid in water (Solvent A) and 0.1% (*v*/*v*) formic acid in acetonitrile (Solvent B). The flow rate was 0.3 mL/min and the gradient was from 80% solvent A: 20% solvent B to 20% solvent A: 80% solvent B within 8 min and then back to 80% solvent A: 20% solvent B within 2 min. Electrospray ionization mass spectra (ESI-MS) were acquired in the positive (ESI+) except for hydrochlorothiazide, which was acquired in the negative (ESI−). LC-MS/MS acquisition parameters for the target molecules were as follows: carbamazepine 237.00 > 178.99 and 237.00 > 194.07; atenolol 267.13 > 145.01 and 267.13 > 190.05; CIP 332.23 > 288.16 and 332.23 > 314.10; ofloxacin 362.24 > 261.11 and 362.24 > 318.17; ERY 734.81 > 158.10 and 734.81 > 576.35; and hydrochlorothiazide 295.93 > 205.00 and 295.93 > 268.96.

The performance of the polymers as sorbents was evaluated by the sorption coefficient, Kd, which is defined as the ratio between the concentration of the drug in solution (Ce) and in the polymer (qe) and it is estimated as the slope of the plot qe (mg/kg) versus Ce (mg/L) at equilibrium.
Kd = dqe/dCe

### 2.5. Modeling of the Sorption Experiments

ISOT-Calc, a macro for MS-Excel, was used to perform non-linear regression to distinct isotherms. The objective function (*U*) was the minimization of the mass balance, which is the difference between the estimated and the experimental Ce values [18]. *U* is defined as the sum of squared residual errors (*e_i_*) obtained from the difference between the experimental and the corresponding values estimated by the guessed model, with *w_i_* being statistical weights:$$U=\sum_{i=1}^{n}w_{i}\cdot e_{i}^{2}$$

Data were fitted to the four parameters isotherm of 2-sites Langmuir, the three parameters isotherms of Vieth–Sladek and Redlich–Peterson, and the two parameters isotherms of Temkin, Freundlich, and Langmuir as defined by ISOT-Calc and depicted in Appendix A. The goodness of fit was evaluated by assessing the standard deviation of the isotherm parameters and the mean weighted square error (*MWSE*) defined as
$$MWSE=\frac{U}{\left(n-p\right)}$$
where *n* indicates the number of experimental points and *p* is the number of refined parameters.

## 3. Results

Sorption is an economically viable wastewater treatment process. The search for sorbent materials is driven not only by their affinity for pollutants but also by the sustainability of their production. We previously reported that it is feasible to obtain sorbent polymers by cross-linking biodegradable carbohydrates with DVS and the ecofriendly material resulting from the cross-linking of starch shows a high affinity for CIP, making it suitable for inland water and seawater remediation [14,15,16]. Nevertheless, this approach failed to yield sorbent materials capable of capturing IBU despite both CIP and IBU showing a certain degree of structural similarity (Appendix A). It was concluded that the cavities formed during the cross-linking were not suitable to accommodate the IBU molecule. On this basis, we hypothesized that the pre-incubation of the carbohydrate with IBU prior to the addition of the cross-linker DVS could preorganize the systems in a manner that resembles the molecular imprinting technology. As it is unlikely that the cross-linking of polysaccharides will generate cavities complementary in size and charge to small molecules such as IBU, we were encouraged by the fact that the cross-linking of cyclodextrin in the presence or absence of toluene yields different polymers, being predominantly linear in the former case and globular in the latter [19].

### 3.1. Synthesis and Characterization

Natural polymers such as chitosan, cyclodextrin, sodium alginate, starch, cellulose, lignin, and their derivatives have been used to prepare molecular imprinted polymers (MIPs) [20]. In addition, along with our previous research, we have generated a library of polymers using starch, Dx, and/or β-cyclodextrin as building blocks [14,15,16]. However, for the purposes of this work, we have focused on Dx because, unlike starch, it exhibits excellent water solubility and, as a linear molecule, it is more flexible than cyclodextrins because the cavities are not preformed. Prior to conducting the cross-linking reaction, the solution of Dx in carbonate was pre-incubated with either 0 mM, 1 mM, or 5 mM of IBU for 5.5 h (Appendix A). Then, DVS was added and the reaction was allowed to proceed overnight with slow agitation, resulting in the insoluble polymers **pDx0**, **pDx1**, and **pDx5**. The isolated yields of the polymers (i.e., percentage of the mass of reactants recovered as an insoluble polymer) were 52.3%, 49.4%, and 33.6%, respectively. The elemental analysis revealed a sulfur content ranging from 5.85% to 6.31% (Appendix A). The Glc/DVS ratio estimated for **pDx0** and **pDx1** was 2.2 and 2.1, respectively, while for **pDx5** it was 2.4. This suggests that the degree of cross-linking is lower at higher concentrations of IBU.

The insoluble polymers underwent characterization using XRPD, FTIR, and SEM. The XRPD analysis (Appendix A) of the starting material Dx revealed dispersive broad peaks centered at 2θ 17° and 22°, indicating the amorphous structure of the materials resulting from its cross-linking with DVS, regardless of the pre-incubation with IBU. FTIR spectra (Figure 1A) display the broad signal of the O-H stretching at 3500 cm^−1^ and the expected double signal at 1282 and 1313 cm^−1^, which are distinctive of the sulfone group and of the DVS cross-linked carbohydrates [16,21]. The spectra of **pDx1** and **pDx5** obtained in the presence of IBU do not show the signature of the C=O stretching, supporting the idea that IBU is not trapped after cross-linking but that it is released during washes in the isolation process, resulting in the synthesis of IBU-free polymers. The polymers underwent further characterization through TGA. Polymers **pDx1** and **pDx5** are very similar but differ slightly from **pDx0**. The latter experiences weight loss to some degree at lower temperatures (Figure 1B). Upon heating the samples to 950 °C in a nitrogen atmosphere at a heating rate of 20 °C/min, a mass loss ranging from 2.7% to 3.9% was observed. The maximum rate of decomposition occurs between 108 °C and 120 °C (Tp1), which is consistent with the expected values for DVS cross-linked carbohydrate polymers. This temperature range is associated with the vaporization of bound water [16]. A second mass loss occurs due to the depolymerization and decomposition of both the polymers and the structure of Dx. The process initiates within the temperature range of 306 °C and 313 °C (T_onset_).

The maximum rate of decomposition (Tp2) is observed at 344 °C for **pDx0** (Appendix A) and 354 °C for **pDx1** and **pDx5** (Appendix A). Organic matter degradation occurs beyond 500 °C, resulting in a residue of ash that accounts for 0.87% to 2.65% of the initial mass. The IR spectra obtained during the analysis reveal signals assigned to CO_2_ (3734, 3626, 2357, 2321, and 666 cm^−1^), CO (2176 and 2116 cm^−1^), and SO_2_ (1375, 1340, 1166, and 1131 cm^−1^). Weak signals were also detected, which may indicate the formation of formaldehyde (2899, 2743, 1749, and 1163 cm^−1^), acetaldehyde (1749 cm^−1^), ethene (3126, 3015, and 948 cm^−1^), and methane (3015 cm^−1^) (Appendix A).

SEM characterization shows that the polymers have a lobular appearance (Figure 2A1–A3), a smooth outer surface, and a homogeneous interior (Figure 2B1–B3). The lobules typically have a diameter in the range of 3 µm, regardless of the concentration of IBU during their synthesis (Figure 2C1,C2). Upon closer inspection, the lobules exhibit a cauliflower-like surface and are connected by filamentous structures (Figure 2C3).

### 3.2. Evaluation of the Polymers as Sorbents of IBU

Providing that a solution of IBU emits fluorescence at 290 nm when it is excited to 260 nm [17], the ability of the polymers to remove IBU from an aqueous solution was evaluated by incubating 0.1 g of the polymer with 10 mL of 10 solutions with concentrations ranging from 0.5 to 8 mg/L (Appendix A). As expected, polymer **pDx0** did not trap IBU from the solution. The pre-incubation and synthesis in the presence of IBU to yield polymers **pDx1** and **pDx5** improved Kd up to 192 L/kg. However, for the purpose of removing pollutants by sorption in WWTPs, sorbents with a Kd lower than 500 L/kg are ineffective [22]. These results led us to conclude that although **pDx1** and **pDx5** differ from **pDx0**, none of them are suitable as scavengers of IBU. We hypothesized that **pDx0**, **pDx1**, and **pDx5** may act as sorbents for other drugs. 

### 3.3. Evaluation of Polymers as Sorbents of a Cocktail of Drugs

The NORMAN network has identified over 700 substances in the aquatic environment of Europe. Among them are the antihypertensives atenolol and hydrochlorothiazide, the antibiotics CIP, ofloxacin, and ERY, and the anticonvulsant carbamazepine [23]. To test our hypothesis, by mass spectrometry we evaluated the polymers as scavengers against a cocktail of these six drugs at concentrations ranging from 50 to 2 µg/L, which falls within the range of concentrations of the emerging pollutants [24]. Table 1 shows that none of the three polymers are effective in removing carbamazepine or hydrochlorothiazide in WWTPs. The removal efficiency of atenolol is close to the 500 L/kg threshold for **pDx0**, while **pDx1** and **pDx5** slightly exceed the threshold for ofloxacin. The use of IBU results in a 14-fold and 10-fold improvement in Kd for the sorption of CIP in **pDx1** and **pDx5**, respectively. The former is suitable for WWTPs, although to a lesser extent than the reported polymer resulting from the cross-linked starch (Kd 1469 L/kg) [16]. This enhanced affinity for CIP was not unexpected, given that CIP and IBU share some degree of structural similarity (Appendix A). However, the exceptional performance of **pDx1** and **pDx5** toward ERY was unexpected. The Kd values for ERY were improved by 26- and 29-fold, resulting in values of 4285 and 4658 L/kg, respectively. This is surprising given that ERY has a larger molecular weight (733.9 g/mol vs. 206.3 g/mol), a higher octanol–water partition coefficient (4.52 vs. 2.17) [25], and higher topological polar surface area (193.91 vs. 32.30 Å^2^) [26] compared to IBU.

### 3.4. Evaluation of the Polymers as Sorbents of ERY

The structure of the aromatic ring of ERY makes it resistant to degradation. More than 5% of the dose of ERY is excreted in urine in the active form and ERY is recalcitrant to various treatments in WWTPs. In fact, ERY has been detected in inland waters, marine systems, groundwater, and, most worryingly, in finished drinking water, calling into question the effectiveness of advanced treatment systems [8]. Therefore, new approaches for the removal and degradation of ERY residues from wastewater are important and sorption is an economically affordable approach. 

The above result of the sorption of ERY from a complex matrix containing five other drugs was pure serendipity and encouraged us to further characterize the sorption of ERY on the IBU-driven pre-organized polymers. First, we evaluated the kinetics of the sorption by mass spectrometry. The analysis of the evolution of the concentration of 1.5 mL of a solution of ERY at 10 µg/L incubated with 15 mg of polymer **pDx1** showed that the equilibrium is reached within 20 min (Figure 3A). The sorption process is very fast and within 2 min of incubation, a remarkable 82.7% of ERY is removed from the solution. In practical terms, this value means that circulating the effluent through a filter or column device for 2 min reduces the concentration of ERY in the outlet water to 1.7 ppb.

Next, the ability of the polymers to remove ERY from an aqueous solution was studied by incubating 15 mg of polymer with 1.5 mL of 12 ERY solutions with concentrations ranging from 2 to 250 µg/L and quantifying the amount of ERY that remains in solution after 30 min by mass spectrometry. The isotherms are very similar, with removal efficiencies close to 93% for **pDx0** and 99% for **pDx1** and **pDx5**. A closer analysis of the isotherms allowed us to estimate Kd as 1.4 × 10^3^, 13.3 × 10^3^, and 6.4 × 10^3^ L/kg for **pDx0**, **pDx1**, and **pDx5**, respectively, which is 2.8, 26.6, and 12.0 times the 500 L/kg threshold for their use as sorbents in WWTPs (Figure 3B). It is important to highlight that the one order of magnitude improvement in Kd for **pDx0** when tested against a solution of ERY has no practical application since with the cocktail of the six drugs the Kd decreases to 161.0 L/kg, showing higher affinity for Atenolol, whereas the improvement for **pDx1** and **pDx5** is not affected by the presence of other pollutants as these do not change the magnitude of their Kd. These data support the high affinity of **pDx1** and **pDx5** for ERY whereas **pDx0** is less selective. 

Using the ISOT_Calc tool [18], the data were fitted to the two-parameter Temkin, Freundlich, and Langmuir isotherms, the three-parameter Vieth–Sladek and Redlich–Peterson isotherms, and the four-parameters 2-sites Langmuir isotherm as defined in Appendix A. Our efforts were focused on non-linear fitting because linearization implies bias, although the linearized forms of the isotherms have been extensively referenced in the literature. Most of the fits converged on a solution for the different isotherms but when the goodness of the fitting was evaluated on the basis of the standard deviation (% r.s.d.) of the parameters defining the isotherm and the mean weighted squared error (MWSE), data only fit to the Temkin isotherm, which assumes that sorption is a multilayer process (Table 2). The values of the constants K_1_ and K_2_ correlate with the Kd estimated for the three polymers. However, they do not provide any additional insight into the sorption mechanism as the Temkin isotherm is an empirical model without effective theoretical support [27].

## 4. Conclusions

Water scarcity is one of the consequences of climate change and emerging pollutants that are resistant to standard water treatments compromise water recycling and reuse, especially in less developed countries where sophisticated approaches are not affordable. Sorption is an attractive approach and the search for sorbent materials is driven not only by the requirement for a Kd greater than 500 L/kg but also by the need for sustainable methods of production. In this context, polymeric materials obtained by cross-linking polysaccharides represent an attractive option, combining both the use of renewable and non-toxic building blocks with their biodegradability. New affinities and improvements in Kd can be achieved by cross-linking in the presence of other molecules that promote pre-organization. This is a versatile and economically viable strategy that can be scaled up, expanding the potential applications of these sustainable sorbents. In particular, the results reported here make **pDx1** and **pDx5** excellent materials for the removal of ERY from contaminated water, with Kd values of 13.3 × 10^3^ and 6.4 × 10^3^ L/kg, respectively, fast sorption kinetics, and good selectivity in the presence of other pollutants

## Figures and Tables

**Figure 1 polymers-16-01090-f001:**
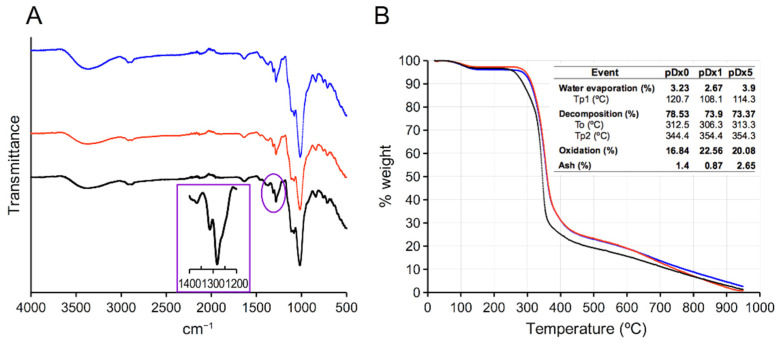
(**A**) FTIR of **pDx0** (black), **pDx1** (red), and **pDx5** (blue). Insert, detail of the 1400–1200 cm^−1^ region showing the double signal at 1282 and 1312 that matches with the distinctive signature of the sulfone group. (**B**) TGA curves of **pDx0** (black), **pDx1** (red), and **pDx5** (blue). Table depicts the significative features, with Tp being the first derivative peak and To being the onset extrapolated temperature.

**Figure 2 polymers-16-01090-f002:**
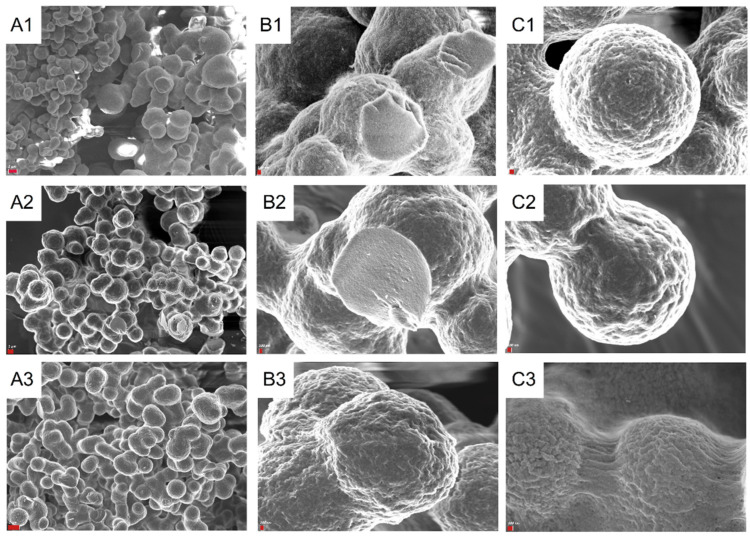
SEM study of **pDx0** (**A1**,**B1**), **pDx5** (**A2**,**B2**,**C2**), and **PDx5** (**A3**,**B3**,**C1**,**C3**). The scale bar (in red) and the magnification are 1 µm and ×2000 for (**A1**,**A2**), 2 µm and ×2500 and for (**A3**), 100 nm and ×14,000 for (**B1**), ×13,000 for (**B2**), ×16,000 for (**B3**), ×19,000 for (**C1**,**C2**), and ×22,000 for (**C3**).

**Figure 3 polymers-16-01090-f003:**
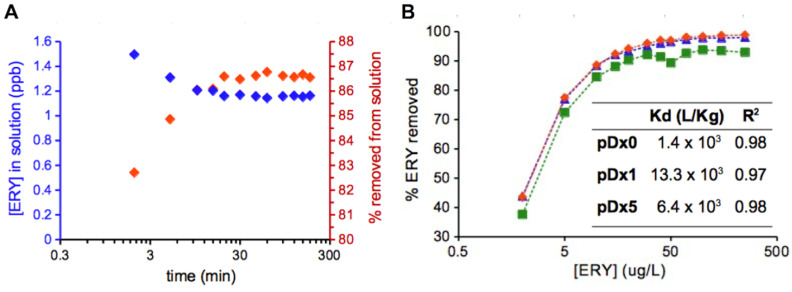
(**A**) Kinetics of the sorption of ERY on **pDx1**. The scale of the X-axis is logarithmic. (**B**) Isotherm of the sorption of ERY on **pDx0** (green), **pDx1** (red), and **pDx5** (blue) expressed as a percentage of ERY removed from the solution. Insert, sorption coefficient (Kd), and coefficient of determination (R^2^). The X-axis is a logarithmic scale.

**Table 1 polymers-16-01090-t001:** Sorption coefficients (Kd) of six drugs analyzed as a cocktail on the sorbents **pDx0**, **pDx1**, and **pDx5**. The coefficients of determination are shown in brackets and italics.

Drug	pDx0	pDx1	pDx5
Atenolol	413.7 (*0.983*)	264.7 (*0.990*)	415.7 (*0.890*)
Hydrochlorothiazide	28.4 (*0.865*)	33.3 (*0.815*)	35.0 (*0.870*)
Ofloxacin	No linear fitting	677.9 (0.915)	631.8 (0.951)
Ciprofloxacin	65.6 (*0.589*)	926.9 (*0.832*)	654.0 (*0.940*)
Carbamazepine	18.0 (*0.989*)	20.1 (*0.968*)	22.5 (*0.990*)
Erythromycin	161.0 (*0.975*)	4285.0 (*0.988*)	4657.7 (*0.954*)

**Table 2 polymers-16-01090-t002:** Values of the parameters and the mean weighted squared error (MWSE) resulting from the fitting to the sorption of ERY to the Temkin isotherm by ISOT_calc. In brackets are the percentage of the root mean square deviation (rmsd).

	K_1_	K_2_	MWSE
**pDx0**	5.951 (1.828)	39.34 (1.944)	0.05
**pDx1**	18.690 (0.908)	128.80 (0.874)	0.01
**pDx5**	13.040 (0.906)	89.180 (0.820)	0.03

## Data Availability

Data are contained within the article and Appendix A.

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
