# Peer review of "Removal of Erythromycin from Water by Ibuprofen-Driven Pre-Organized Divinyl Sulfone Cross-Linked Dextrin"

_polymers, 2024, doi:10.3390/polym16081090_

Round 1

Reviewer 1 Report

Comments and Suggestions for Authors

In the current work, the authors investigated the removal of Erythromycin from impacted water by a novel sorbent polymer derived from cross-linking of dextrin with divinyl sulfone (DVS). SEM analysis was used to characterize the prepared materials. Then the efficiency of the sorbents in removing different contaminants, including ibuprofen, a mix of six drugs, and erythromycin. The study is quite interesting but there are some concerns in this work that I would like to introduce to the authors before publication. Authors should consider these suggestions positively, and needs to be addressed properly, in order to meet the standard of the journal. Recommendation: Minor Revisions

1) [Manuscript] There are some grammatical errors throughout the manuscript and the overall reading is not smooth. Please, re-proofread and fix the whole MS to avoid grammatical issues. 

2) [Abstract] Rather than just summarizing the activities carried out in the study, please consider listing more results already here in the abstract section. The readers should read the main information, benefits, drawbacks, and outcomes in the abstract.

3) [Introduction] The authors lack to highlight the novelty in the work. I would suggest adding more information regarding the hypothesis of this work. The introduction should make it clear, why this particular study is necessary, and in which lies its novelty. Authors should emphasize the relevance, uniqueness, scientific novelty, and technological and environmental significance of this particular study compared to previous similar studies. The research problem(s) addressed in this study should be clearly formulated in the context of bridging knowledge gaps from previous studies.

4) [Introduction] While discussing the alternatives of ERY and other antibiotics treatment, the option of using advanced oxidation processes technology is worth of mentioning. Please see https://doi.org/10.1016/j.scitotenv.2019.135023; https://doi.org/10.1016/j.cscee.2023.100433.

5) Moreover, why did the Authors finally decide to use adsorption to remove antibiotics from impacted water? What are the limitations of alternative technologies? And what are the benefits of adsorption? Please consider providing a more thorough rationale for your choice.

6) [Materials and Methods] Authors are also kindly asked to briefly describe how data quality control was assured.

7) [2.4 Sorption studies] The authors please provide a suitable reference on the method chosen for analyzing the cocktail of drugs by UPLC- mass spectrometry

8) [Manuscript] The present study missing the comparison of this work with the other ERY adsorption reports in terms of removal efficiency, adsorbent materials, operative conditions, etc.

Comments on the Quality of English Language

There are some grammatical errors throughout the manuscript and the overall reading is not smooth. Please, re-proofread and fix the whole MS to avoid grammatical issues

Author Response

We thank the referee for his/her comments that have contributed to improving our manuscript. They have been addressed as follows (changes in the manuscript are shown in red):

Answer to comment 1 regarding the grammatical errors.

We realized that some typos came from the different versions of the software used to edit the manuscript.

Action taken: The current version has been thoroughly revised, double checked by a native English speaker, and re-proofread when upload to MDPI before submission.

Answer to comment 2 regarding the abstract

Action taken: The abstract has been modified to include the selectivity of the materials toward ERY and the fast kinetics of the sorption, and to highlight the values of Kd toward ERY in the context of the 500 L/kg threshold for their practical implementation, as well as their potential use in WWTPs.

Answer to comment 3 regarding modification the introduction section to highlight the novelty of the work, provide additional information on the hypothesis, contextualize and emphasize its relevance

Action taken: As suggested by the referee, the following paragraph has been included at the end of the introduction section (lines 84-90):

In this work we further explore our hypothesis of using DVS cross-linked carbohydrates to obtain affordable insoluble sorbent materials as an alternative to more sophisticated approaches to treat wastewater. We report that the cross-linking of Dx in the presence of IBU yields polymers with a high affinity toward ERY. This strategy based on cross-linking in the presence of other molecules that promote a pre-organization to improve Kd and yield new affinities expands the potential application of cross-linked carbohydrates sorbent polymers to different pollutants

Answer to comment 4 regarding the mention of the advanced oxidation process technology introduction section

The lack of mention of the advanced oxidation technology is a consequence of an error during the process of editing to fit the manuscript to the template. In fact, reference [13] is a review compiling the different methods currently available for the removal of antibiotics from water and wastewater and, including advanced oxidation technology.

Action taken: the advanced oxidation technology has been included in the introduction section among the methods. The corrected sentence is the following (lines 72-75):

The removal of ERY from water has been addressed by the implementation of various methods including coagulation-flocculation, powered activated carbon, granular activated carbon, reverse osmosis, advanced oxidation processes, membrane bioreactors or biological activated filters, with varying degree of success [13]

Answer to comment 5 regarding the benefits of adsorption and the limitation of the other technologies

When we started to work on the removal of pollutants from wastewater, we were aware that processes that work in the lab may not be real-life solutions. However, we did not realize the magnitude of the task until we visited the Biofactoria Sur in Granada and exchanged ideas with the engineers. The depuration process involves the circulation of about 50000 m3/day of wastewater along different tanks, the recycling of sludge as compost, and the use of biogas to generate energy.

The use and management of insoluble sorbent materials such as the DVS cross-linked carbohydrate does not represent a challenge for the WWTPs already in use. The contact between the wastewater and the polymers may take place in one of the tanks, the sorption is very fast, and it does not imply a long delay in the workflow. The separation is just a matter of decantation, and the polymer, once saturated, can be reused after desorption, recycled as compost, or degraded by bacteria to produce biogas. On the other hand, Biofactoria Sur is located downstream of a hospital that is a source of antibiotic pollution. The pre-treatment of the wastewater from the hospital would represent a clear strategy to facilitate the task of Biofactoria Sur.

In this context, the use of DVS cross-linked carbohydrate does not require important investments or building works and can be implemented in both WWTP or upstream of the discharge point of problematic activities. These are favorable features compared to the other technologies, especially when considering less developed countries.

Action taken: none since the scope of our current research is focused on basic science and the incorporation of the above discussion would require to scale up our studies and shift our research to applied science.

Answer to comment 6 regarding data quality control

The sorption assays were performed several times during the optimization, and the reported data were gathered in triplicate experiments.

Action taken: the first sentence of the subsection “2.4 Sorption experiments” (line 143) has been modified as follows:

Sorption experiments were conducted in triplicate in batch mode at room temperature

Answer to comment 7 regarding the bibliography supporting the method chosen for the analysis of the cocktails of drugs by UPLC-mass spectrometry

The use of UPLC-MS of the analysis is a well-established approach for the detection and quantification of emerging pollutants.

Action taken: the following reference has been included in as [24] (lines 457-459) and the cites have been renumbered accordingly:

Perez-Fernandez, V.; Mainero Rocca, L.; Tomai, P.; Fanali, S.; Gentili, A. Recent advancements and future trends in environmental analysis: Sample preparation, liquid chromatography and mass spectrometry. Anal. Chim. Acta 2017, 983, 9-41, doi: 10.1016/j.aca.2017.06.029.

Reviewer 2 Report

Comments and Suggestions for Authors

Notes

Introduction

The Introduction section needs to be promoted by adding over the past two years articles devoted to polymer sorbents and various pollutants (see the works of Yudaev P. A. et al.). What are the features of the antibiotic sorption process in comparison with reverse osmosis, membrane separations, coagulation-flocculation and other processes?

Materials and methods

An adequate table should be provided indicating the loadings of the starting substances for obtaining sorbent samples (to make the text easier to read).

results

A diagram of the dextrin cross-linking regime and a diagram of drug sorption should be given.

It is necessary to provide the IR spectra of cross-linked dextrin, internal combustion engine, sorbent, saturated drug, mainly the text of the article.

The difference in Kd results for different drugs and sorption mechanisms should be explained.

conclusions

Where exactly should the sorbent be used? What research will be done in the future? How will the authors desorb ERY from the sorbent, will the sorbent work over several sorption/desorption cycles?

Author Response

We thank the referee for his/her comments that have contributed to improving our manuscript. They have been addressed as follows (changes in the manuscript are shown in red):

Answer to comment regarding the need to add articles devoted to polymer sorbents over the past two years and the recommendation to check the papers published by Yudaev P.A.

We have searched for Yudaev at ORCID and found the following ORCID IDS that may match with Yudaev, P. A.:

0000-0002-4356-4426: Yudaev, Pavel (Web of Science ResearchID: N-3283-2015)

0000-0001-8562-4696: Yudaev, Pavel (Scopus Author ID 572053244339)

Since this author has published 17 articles and the closest related to the topic of our work is: “Gel based modified chitosan for oil spill cleanup” (J. Appl. Pol. Sci. 2024, 141(4), 254838), we concluded that this is not the Yudaev that referee 2 suggested to check. Unfortunately, with the information provided by the referee, we have been unable to find the author Yudaev, P.A.

The bibliography reporting the use of sorbents to remove antibiotics from water is very abundant and is the topic for a review. In addition, the comparison of the sorption process with reverse osmosis, membrane separation… is out of the scope of our work which is focused on the cross-linking of dextrin in the presence of IBU to improve Kd and yield a higher affinity and selectivity toward ERY, expanding the potential application of the cross-linked carbohydrates sorbent polymers to different pollutants.

Action taken: none

Answer to comment regarding the inclusion of a table in materials and methods depicting the reagents and amounts used to obtain the sorbent polymers

The description of the synthesis has been improved, and chemists should be able to reproduce it.

Action taken: as suggested by the referee and bearing in mind that the readers of Polymers come from different disciplines, for the sake of clarity, a table indicating the loadings of the starting substances for obtaining the sorbents has been included as supporting material and cited in the manuscript as Table S1 (line 116).

Answer to comment regarding the inclusion of a diagram of the dextrin cross-linking regime and a diagram of the sorption

It is not clear to us what the referee means by “cross-linking regime”. The cross-linking is a Michael addition of the alkoxy groups resulting from the hydroxy groups of the dextrin to the vinyl sulfone groups of the DVS.

The sorption process is a standard procedure that consists of contacting the polluted water solution with a defined amount of polymer for a certain time.

Action taken: since it is not clear to us what the referee suggests by a diagram of the dextrin cross-linking regime and the sorption process is a standard procedure, no action has been taken.

Answer to comment on the presentation of the IR spectra of the cross-linked dextrin, internal combustion engine, sorbent, saturated drug

The IR spectra, TGAs curves, derivative TGAs, and IR-TGAs were included as supported information to save space. However, we do not understand what the referee intends to mean by internal combustion engine, sorbent, saturated drug

Action taken: As suggested by the referee, IR spectra and TGAs have been moved to the main text as a new Figure 1 and the other figures have been renumbered accordingly.

Answer to comment regarding the differences in Kd for different drugs and the sorption mechanism

We have not addressed the study of the mechanism of the sorption of the drugs, and this is far from trivial due the range of concentrations assayed. In a previous work at a different concentration scale, we conducted experiments to gain insight into the sorption of ciprofloxacin (CIP) on cross-linked starch (doi 10.3390/polym15153188). Results led us to hypothesize that the main driving force is the ionic interaction between the carboxylate group of CIP− and polymer and, secondarily, a weaker interaction with the lone electron pair of the nitrogen in the piperazine ring linked to the quinolone.

The Kd values are related to affinity. The higher Kd, the higher the affinity of the pollutant toward the sorbent. As discussed in the article, the polymer obtained by cross-linking in the absence of IBU shows an ERY Kd of 1.4x103 L/Kg that drops to 161 L/Kg when assayed against the cocktail of drugs where the Kd of the polymers obtained in the presence of IBU remains in the same range.

Answer to comment regarding where the sorbent should be used and our future research

The depuration process involves the circulation of large volumes of wastewater along different tanks and the sorbent can be incorporated either as an independent final treatment in a specific tank or pipe or as part of the primary treatment. In any case, the use and management of insoluble sorbent materials such as the DVS cross-linked carbohydrate does not represent a challenge for WWTPs already in use. In addition, hospitals are well known sources of antibiotic pollution. The pre-treatment of the wastewater from the hospital before discharge would represent a feasible strategy to facilitate the task of WWTPs.

Our future research will be focus on synthesizing new insoluble polymers by cross-linking in the presence of other molecules that promote a pre-organization to improve Kd and yield new affinities, in order to expand the potential application of the cross-linked carbohydrates sorbent polymers.

Answer to comment regarding the regeneration and number of sorption/desorption cycles

Along our research, we have reused the polymers several times and we have not noticed a poorer performance.

Round 2

Reviewer 1 Report

Comments and Suggestions for Authors

The authors responded to all reviewer comments. The manuscript is ready to be published now.

Author Response

We thank the referee for granting the publication of our work. His/her comments have contributed to improve our manuscript.

Reviewer 2 Report

Comments and Suggestions for Authors

I apologize for any difficulties encountered while reading the review.

Please add to the Introduction section an article devoted to a biodegradable sorbent based on the natural biodegradable polymer chitosan to expand the current state of research in the field of sorption of various pollutants from water bodies (see work https://doi.org/10.1002/app.54838).

I meant that it was necessary to provide a figure to show the cross-linking reaction of dextrin.

Sorption data for several cycles should be provided.

A figure of the interaction of the sorbent with erythromycin should be provided.

Author Response

We thank the referee for his/her comments. They have been addressed as follows (changes in the manuscript are shown in red):

Answer to comment regarding the need to add the article DOI 10.1002/app.54838 by P. Yudaev, A. Semenova and E. Chistyaknow

As already discussed in our previous reply the article “Gel based modified chitosan for oil spill cleanup” (J. Appl. Pol. Sci. 2024, 141(4), 254838) is out of the focus of our research. Yudaev et al reported the synthesis of a gel by cross-linking with glutaraldehyde and reaction with citral as hydrophobic moieties responsible for the interaction with the oil waste to be removed. Important differences with our research are:

1. Chemistry: the Yudaev´s approach is based on the formation of a Schiff base (imine) between the amino group of chitosan and the carbonyl groups of either glutaraldehyde or citral, whereas our approach relies on oxa-Michael addition of the alkoxy groups resulting from the deprototanion of the hydroxy groups of the dextrin on the Michael acceptor divinyl sulfone (DVS). In addition to the chemistry, an important difference is the stability. Whereas Schiff bases are reversible (doi 10.1002/macp.201800484) and their formation has often been considered for the synthesis of self healing hydrogels (doi 10.3390/molecules24163005). The reaction of alkoxy group yields an ether that is stable (the synthesis was carried out at pH 12 and retro-Michael would imply even higher pH values).

2. Target molecules to be removed. We have focused on a very specific and persistent problem: anthropogenic pollution of water with antibiotics. Yudaev´s work is focused on (accidental) spills of oil. Thus, in our case, the sorption is evaluated in the range of concentrations of emerging pollutants (ppb) which is orders of magnitude lower than those assayed by Yudaev

Action taken: none because the only common feature of both studies is the use of carbohydrates as the starting material for cross-linking. If Yudaev´s work were included in the introduction, all the works reporting cross-linked carbohydrates should also be cited and this is out of the scope of our manuscript.

Answer to comment regarding the inclusion of a figure showing the cross-linking reaction of the dextrin

We apologize for not understanding the previous comment on this issue.

Action taken: A new figure depicting the synthesis has been included as Scheme S1 in Supplementary Material and cited in line 218 of the manuscript.

Answer to comment regarding data for several cycles of sorption

As already stated in our previous comment addressing this issue, the polymers have been used several times without any appreciable variation in their performance. (please, recall that we are working in a ppb range). No study was specifically addressed because, since we evaluated the polymers against a cocktails of drugs with different features, to assure their removal and the regeneration of the polymers, they were washed with the following polarity range: water > methanol > ethyl acetate > hexane and finally with ether to dry the polymer. In practice, because these polymers trap different substances, any protocol of regeneration set up in the laboratory will be of poor application in a complex matrix such as wastewater. Hence, no additional efforts were made.

Action taken: None

Answer to comment on including a figure of the interaction of the sorbent with ERY

Erythromycin is a complex molecule and its interaction with the sorbent is far from trivial, and we do not have a reasonable hypothesis supported by experimental data to describe the interaction. Just to illustrate the situation , consider the particular case of cyclodextrin (cyclic oligosaccharides de 6, 7 or 8 glucose). They arrange creating a cone shape with an interior cavity that can host hydrophobic molecules whereas the outer surface of the molecule is hydrophilic. Please, recall that dextrin is much longer, that was pre-organized by incubation with ibuprofen and that the cross-linking does not yield simple cyclic molecules but complex 3D lattices.

Action taken: None